# Do patients' characteristics influence their healthcare concerns?—A hospital care survey

**George G. A. Pujalte, Isaac I. Effiong, Tais G. O. Bertasi** **\*, Raphael A. O. Bertasi, Susannah S. Rothstein, Ryan Cudahy, Lorenzo O. Hernandez, Timothy M. Davlantes**

Department of Family Medicine, Mayo Clinic, Jacksonville, Florida, United States of America

\* taisg12@me.com

## Abstract

### Background

Hospital performance is often monitored by surveys that assess patient experiences with hospital care. Certain patient characteristics may shape how some aspects of hospital care are viewed and reported on surveys.

### Objective

The aim of the study was to examine factors considered important to patients and determine whether there were differences in answers based on age, gender, or educational level.

### Methods

Cross-sectional study based on a hospital survey developed via literature review and specialist recommendations. This study included randomly selected patients 18 years or older who were recently admitted to the hospital or admitted more than 50 days before the survey was being applied. Survey domains included age, gender, educational level, factors considered important for the health care in a hospital setting and sources of information about hospital quality used by each subject. Answers description and statistical analysis using Fisher exact test were performed.

### Results

The survey was applied to 262 patients who were admitted under different services. The most important concern reported was the risk of getting a hospital-acquired infection (67.18%), followed by understanding explanation from the doctors' plans (64.12%) and doctors' ability to listen carefully (58.78%). Women are more concerned about their risk of falling ($p = 0.03$). Patients older than 65 years find important that the doctors explain everything in a way they can easily understand ($p = 0.02$), while lower educated patients consider most if the doctor treats them with courtesy and respect ($p = 0.0027$).

### Conclusion

Patient characteristics have an effect on how hospital care is perceived. Regardless of the characteristics of the population, the risk of getting an infection was the main concern

**Data Availability Statement:** All relevant data are within the manuscript and its Supporting Information files.

**Funding:** The authors received no specific funding for this work.

**Competing interests:** The authors have declared that no competing interests exist.

overall, so it is important that hospitals promote actions to prevent it and share them with patients.

## Introduction

Standardized survey instruments and data collection methodologies are often used to measure and publicly report patients' assessments of hospital care. The Agency for Healthcare Research and Quality and the US Centers for Medicare & Medicaid Services monitor such surveys and may publicly report hospital-level results [1]. Hospital surveys use standardized questions and administration protocols, permitting an assessment of patients' experiences of hospital care as well as the ability to monitor changes in hospital performance over time [2].

The main purposes of hospital surveys are to facilitate objective and meaningful comparisons of hospitals on topics that consumers deem important, create incentives for hospitals to improve quality of care, and enhance public accountability in health care by increasing the availability and transparency of information [3]. Topics that patients deem important may vary from hospital to hospital. These topics deserve further study as they have the potential to guide policies and care characteristics. Stakeholders across the health care spectrum consider hospital metrics to be very important markers of the quality of care provided in hospitals [4]. Consumer groups, health care providers, employers, and state and federal governments have made measuring and improving hospital quality of care top priorities [4].

Some factors that affect patient survey scores may not be directly related to hospital performance [5]. Even as hospitals strive to treat everyone equally and surveys are designed to capture responses that would be representative of the overall population, administrators may find it useful to understand why their hospitals are getting specific score trends. The latter may be linked to inherent variabilities in the population that is within the catchment of their hospitals. Education level and age of patients could be factors, for example, in that educated and younger patients may tend to evaluate health care less positively [6, 7]. It is important to understand which aspects of a survey are regarded as essential to patient care by the patient population being surveyed. There may be discordance between what is being surveyed and what is relevant to specific patient populations, and care may be patterned to address what patient populations deem important. The degree to which patients feel respected or have their needs responded to are examples of patient-centered measures. Patients are often the only or best source for such data, which can be generated using standardized, well-developed experience measures that complement measures for technical care quality [8]. Earlier studies have suggested that specific care experiences, such as whether nurses and doctors listened carefully, affected which aspects of hospital surveys are important to survey participants [9, 10].

Standardized surveys are a great instrument to assess the hospital quality. However, further investigation is needed to understand the patients' concerns behind each response to these surveys. Therefore, the primary aim of this study was to examine hospital care factors considered important to patients and to determine whether there were differences in answers based on age, gender, or education. This study has the potential to allow administrators to adjust aspects of care to reflect what patients in this setting deem to be essential.

## Methods

This cross-sectional study was approved by the Institutional Review Board (16–000546). A survey developed via literature review and specialist recommendations was applied in different

departments of the tertiary care center where this study was done, during the year of 2016 [11, 12].

Patients were eligible to participate in the study if aged 18 years or older, admitted to the hospital no more than 50 days before data collection, being able to understand the study and to fill out the survey on paper. The survey was conducted personally by six authors of this study in the waiting rooms of all hospital buildings and in patients' rooms, and in hospital rooms, depending on the setting they are in, at different days of the study period. The interviewers approached the patients randomly as they were coming out of each room, explained the objectives of the survey and the strictly confidential treatment the information would receive. Then, they invited each patient to participate in the survey. A written informed consent of the participants was obtained before collecting any data. The response rate was 64.8%.

By using a paper questionnaire, data was collected regarding demographic information (age, gender, educational level, ethnicity, Spanish, Hispanic or Latino origin, main language spoke at home) and in which department the survey was conducted. The surveys were completely anonymously with no patient identifiable details.

The participants were requested to answer thirteen questions according to the command. For the first nine questions regarding factors considered important for the health care in a hospital setting, the answers should be chosen between "very important" or "not important". The latter questions have specific answers options regarding sources of information about hospital quality used by each subject. The survey form template can be accessed in the Appendix (Supplementary material). Finally, the participants were asked to choose from the first nine questions, which of them were considered the most important for them, being able to select more than one for this step.

Descriptive analysis of survey answers is reported as frequency and percentage. In order to compare the answers, participants were divided in 3 subgroups by age (>65 years old and <65years old), gender (men and women) and educational level (<4 years of college and ≥4 years of college) and analyzed using Fisher exact test. Statistical analysis was performed with SPSS (version 1.0.0.1347) for Mac OS. All statistical tests were 2-sided with the alpha level set at .05 for statistical significance.

## Results

The survey was answered by 262 patients, comprised of 129 (49.24%) men and 133 (50.76%) women with a median age of 70 years, ranging from 20 to 95 years (mean 67.76 ± 14.54). The majority of the participants described themselves as White (230; 87.79%) with no Spanish, Hispanic or Latino origin (247; 94.27%) and with main language spoke at home being English (257; 98.09%)–Table 1. At least four years of college was completed by 135 participants (51.53%).

The department with more participants was Family Medicine (106; 40.46%) followed by Internal Medicine (90; 34.35%)—Fig 1.

Overall, the risk of getting a hospital-acquired infection was considered the most important concern of patients (176; 67.18%), followed by an understandable explanation of the doctors' plan (168; 64.12%), and doctors' ability to listen carefully (154; 58.78%). The risk of falling while in the hospital (40; 15.27%) and chance of returning to the hospital after discharge (78; 29.77%) were considered the two least important concerns among the other questions (Figs 2–4).

Only 49 patients (18.7%) searched information about the hospital on the internet before going to an appointment—24 used only one search tool while 25 used at least two. Google was the most used in overall (51.02%) and as a single tool, followed by the US News and World Report (38.78%) and Healthgrades (36.73%)–Fig 5.

Table 1. Demographic information (N = 262).

| Age | N (%) |
|---|---|
| Less than 65 years old | 93 (35.5) |
| More than 65 years old | 169 (64.5) |
| **Sex** | |
| Men | 129 (49.2) |
| Women | 133 (50.7) |
| **Ethnic origin or descent** | |
| Not Spanish/Hispanic/Latino | 247 (94.3) |
| Puerto Rican | 5 (1.9) |
| Mexican/Mexican American/Chicano | 2 (0.8) |
| Cuban | 1 (0.4) |
| Other Spanish/Hispanic/Latino | 3 (1.15) |
| Missing | 4 (1.5) |
| **Main language** | |
| English | 256 (97.7) |
| Spanish | 1 (0.4) |
| Chinese | 1 (0.4) |
| English + Spanish | 1 (0.4) |
| Other | 3 (1.1) |
| **Educational level** | |
| Some high school, but did not graduate | 6 (2.3) |
| High school graduate or GED | 34 (13) |
| Some college or 2-year degree | 87 (33.2) |
| 4-year college graduate | 64 (24.4) |
| More than 4-year college degree | 71 (27.1) |
| **Ethnicity** | |
| White | 230 (87.8) |
| Black or African American | 22 (8.4) |
| Asian | 6 (2.3) |
| American Indian or Alaska Native | 1 (0.4) |
| Missing | 3 (1.1) |

At the end of the questionnaire, the participants were asked about their preference in how to view the measures of hospital improvement. The most preferable was a list (64.12%) followed by bar chart (39.69%), line graph (18.32%) and pie chart (14.50%).

Survey answers were compared among the three subgroups (age, gender and educational level). Although in overall the risk of falling while in the hospital was considered one of the least important, when comparing it by gender, 27 (20.3%) women considered it more important compared to 13 (10.08%) men, which was statistically significant ($p = 0.03$)–Fig 2.

Patients older than 65 years (117; 69.23%) were more concerned if the doctors explained things in a way that they could understand than younger ones (51; 54.84%) ($p = 0.02$)–Fig 3.

The respectful doctor's treatment was considered more important for participants with lower (<4 years of college) than greater educational level (≥4 years of college)– 67 patients (52.76%) vs 46 (34.07%), $p = 0.0027$, respectively–Fig 4.

The way to view measurements of hospital improvement was compared: men have a preference to line graphs (30; 23.26%) compared to women (18; 13.53%)–$p = 0.042$, while participants with higher educational level (≥4 years of college) preferred bar (34; 25.19%) and line (63; 46,67%) charts compared to those with lower educational levels (14; 11.02% and 41;

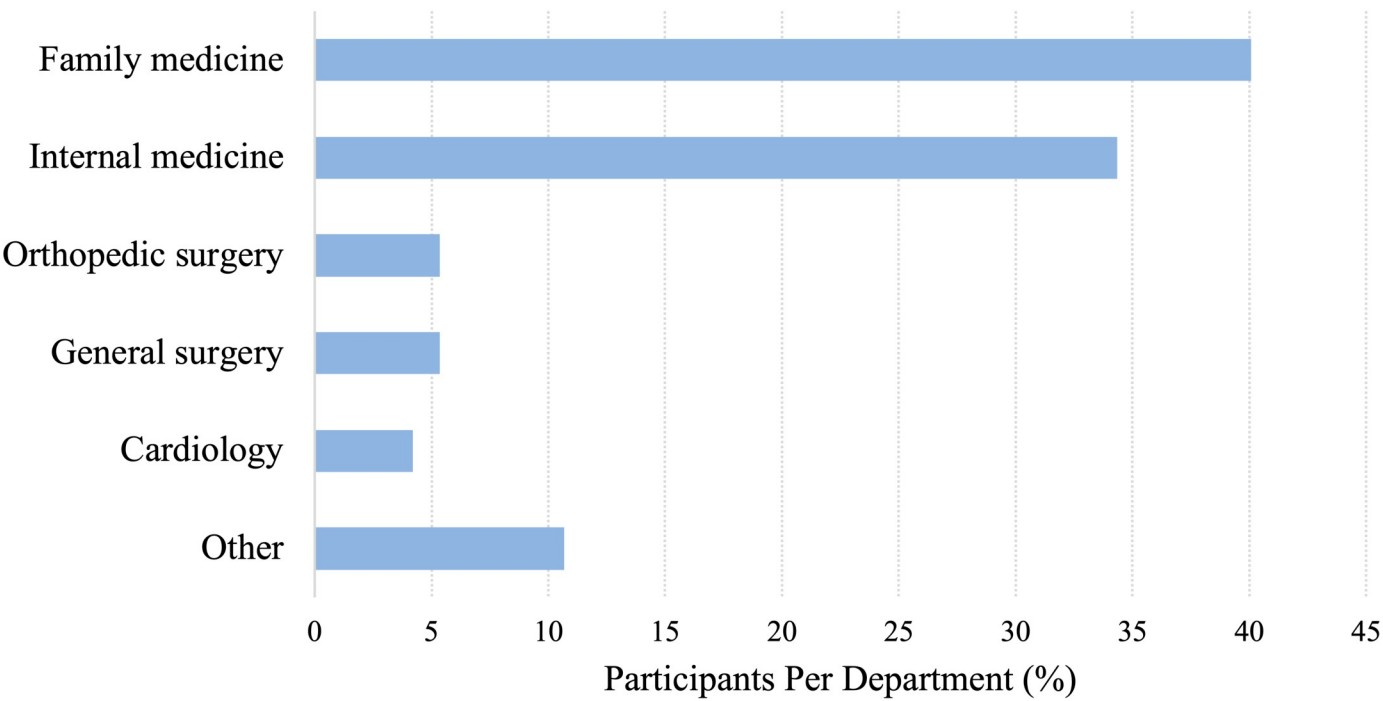

**Fig 1. Participant frequency according to department (N = 262).**

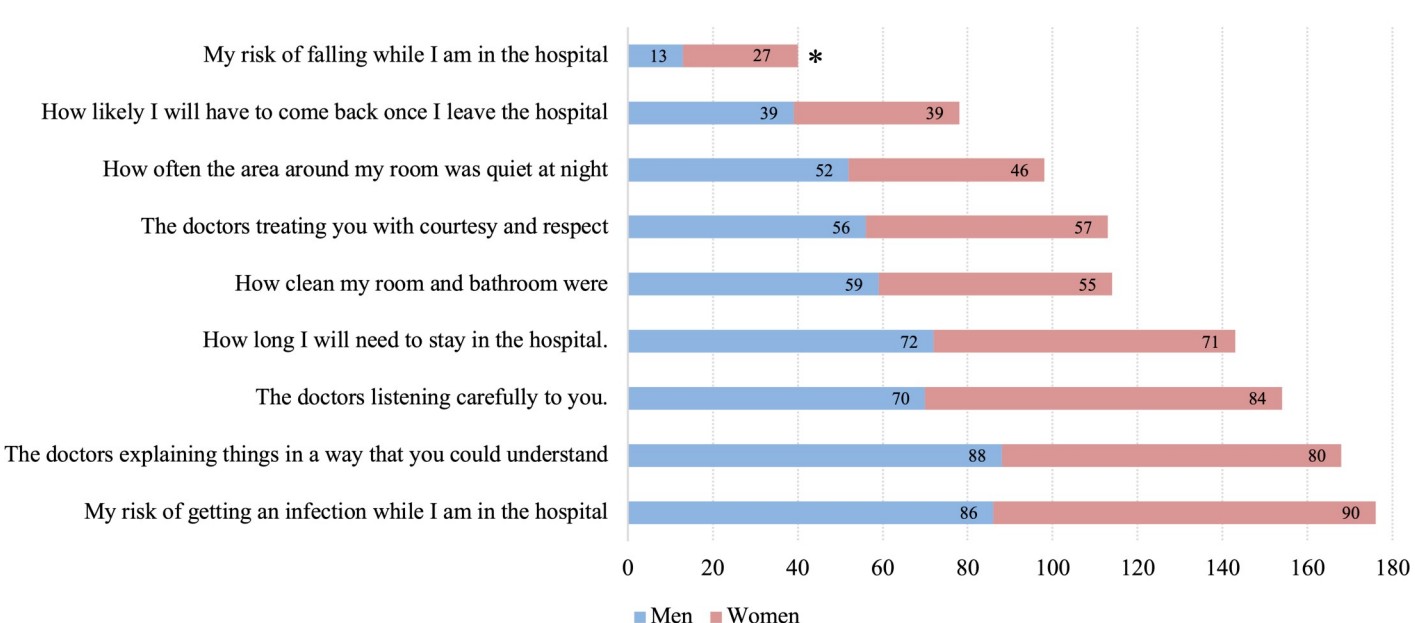

**Fig 2. Survey answers according to sex.** Frequency of participants that answered each question as "Very Important" in the hospital care, according to sex. *Statistically significant.

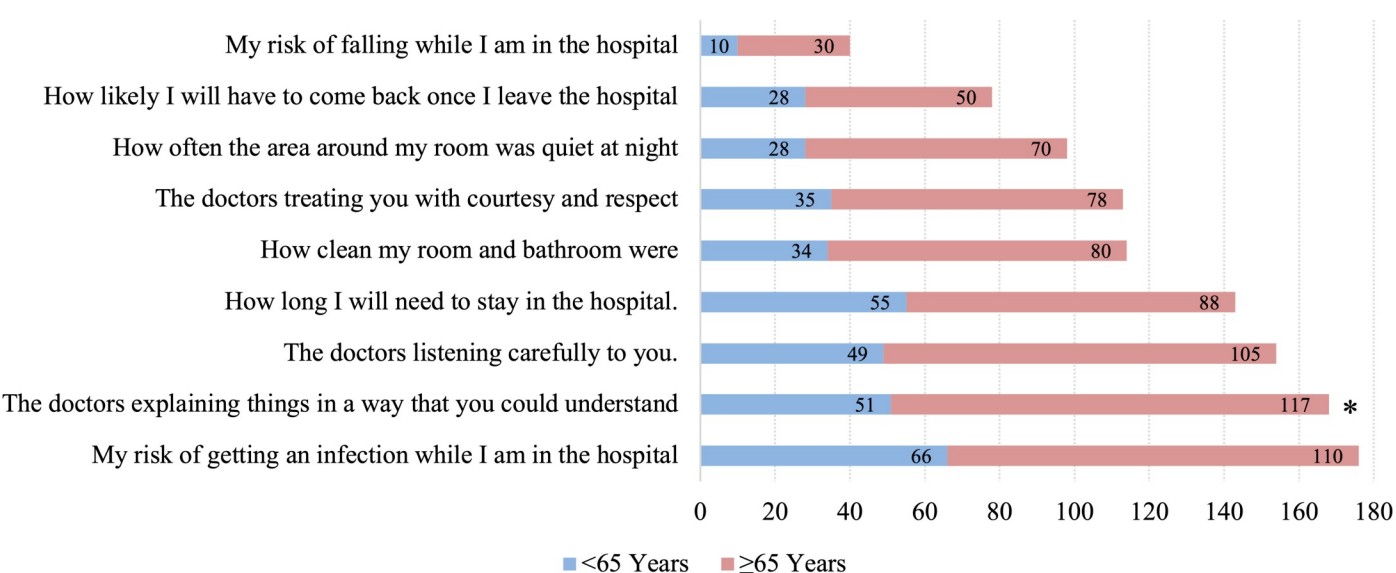

**Fig 3. Survey answers according to age.** Frequency of participants that answered each question as "Very Important" in the hospital care, according to age. * Statistically significant.

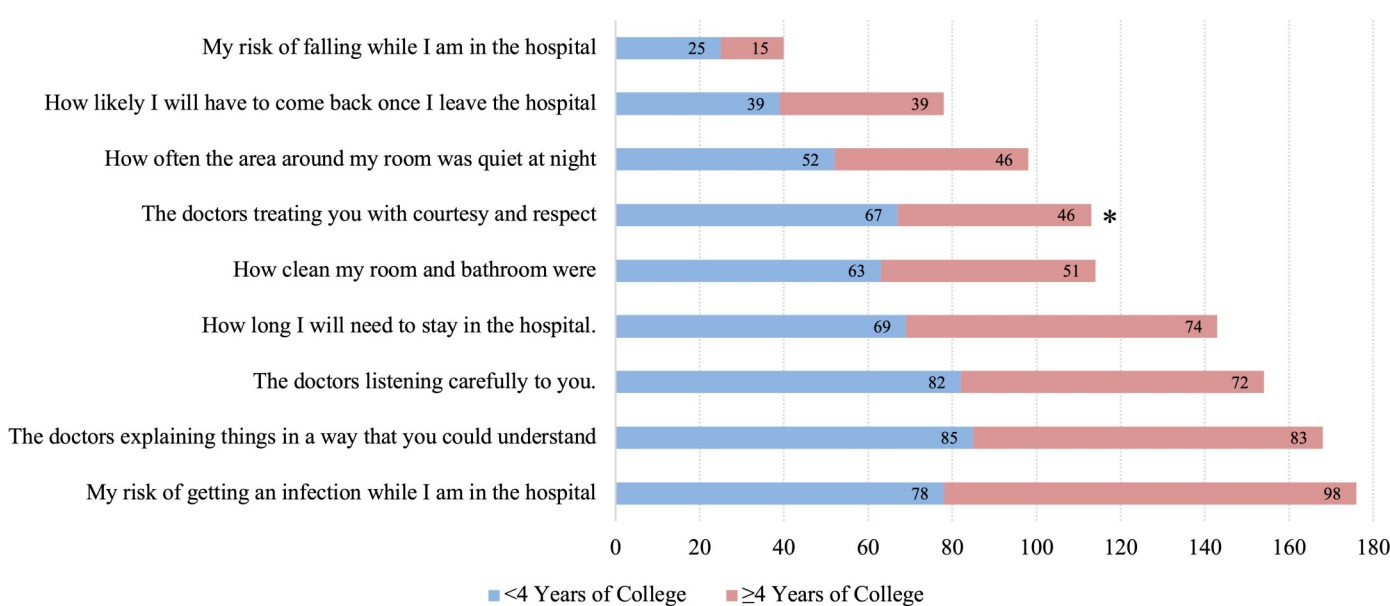

**Fig 4. Survey answers according to educational level.** Frequency of participants that answered each question as "Very Important" in the hospital care, according to educational level. *Statistically significant.

**Search Tools (N=49)**

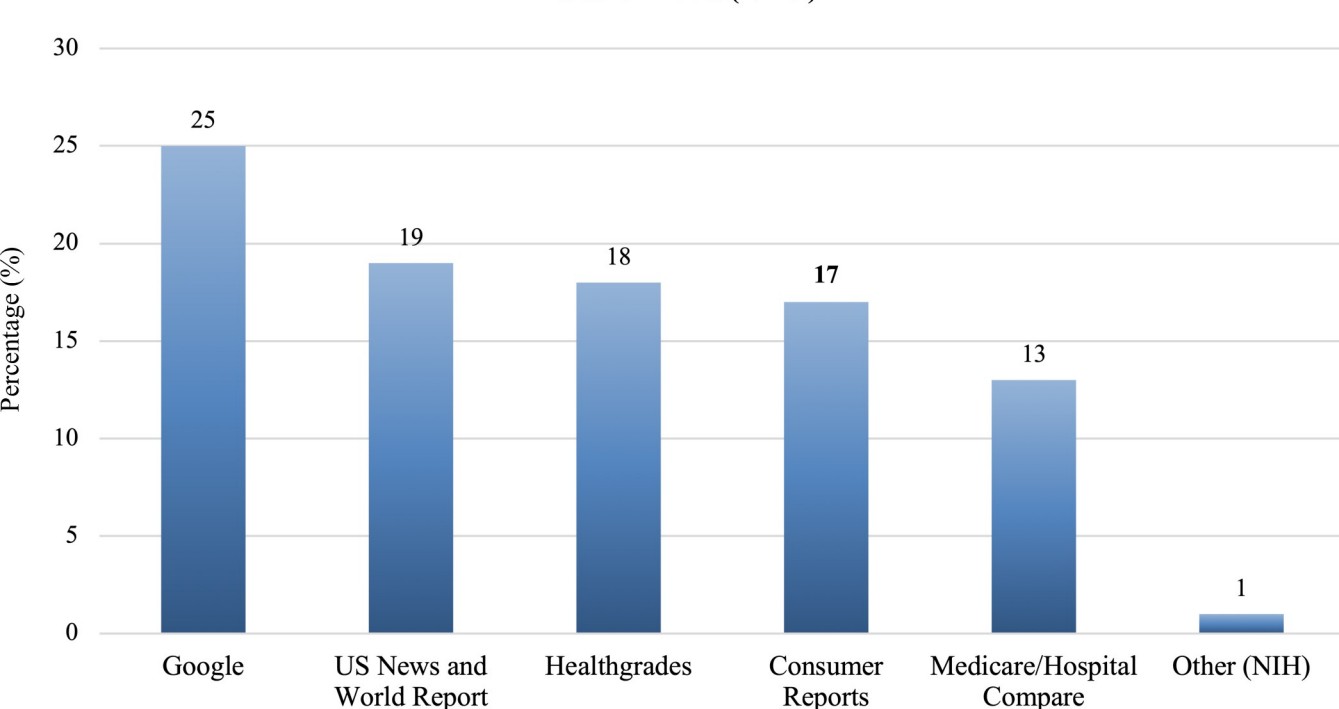

**Fig 5. Search tools (%).** Frequency of search engine tools (N = 49). Twenty-five patients chose more than one tool.

32.28%)–p-values **0.03** and **0.017**, respectively. Although it was not statistically significant comparing within subgroups, the most preferred view was as a list, regardless of the gender and educational level.

In regards to which tool was used to search information of the hospital, comparing their respective subgroups, participants younger than 65 years old used more Healthgrades® (**p = 0.19**) and US News and World Report® (**p = 0.09**), while participants with higher educational levels chose Medicare/Hospital Compare® (**p = 0.14**). There were no other statistical differences between the subgroups for other questions (**p>0.05**).

## Discussion

This study revealed that getting an infection was the most important concern (67.18%) reported by patients. Acquiring an infection while receiving health care is also known as "health care-associated Infections" (HAI) [13]. Although their incidence have been decreasing in the last few years [14], particularly due to measures to prevent urinary tract and surgical site infections, at least one in 31 hospitalized patients (3.2%) develops HAI every day [15]. Perhaps, due to the HAI's high incidence, it is a primary concern for patients. The US Center for Disease Control and Prevention reported that nearly 1.7 million hospitalized patients acquire HCAIs while being treated for other health issues and that more than 98,000 patients (one in 17) die due to these, every year [16]. This concern may be also associated to the fact that HCAIs are frequently reported in the media happening to known people and patients may consider their chance of getting an infection while being in the hospital directly related to other things they care about also, such as cleanliness of the hospital, level of care that the nurses give, and appropriateness of medications, such as antibiotics. Therefore, evidence-based prevention strategies adopted by health care settings, such as hand hygiene, early catheter

removal, and reduction of unnecessary antibiotic prescription [17], could be shared with the patients in attempts to address their concerns regarding getting an infection.

The following two most important concerns reported in this study were regarding the doctors' ability to properly explain a therapeutic plan (64.12%) and to listen carefully to the patients (58.78%). Effective doctor-patient communication is essential in delivering high-quality health care, since more accurate information is necessary for diagnosing, adherence to therapeutic plans, decreased length of hospital stay, and many other positive factors that impact the clinical outcomes are associated to patients' reports of good doctor-patient communication [18]. However, considering communication as satisfactory is not always mutual. A survey with 807 patients and 700 orthopedic surgeons showed a discrepancy in the perception of a good communication as more doctors (75%) tended to be satisfied compared to patients (21%) [19].

Many factors justified the barriers for good communications such as a high amount of doctors' workload, patient's anxiety and unrealistic expectations, doctor's avoidance behavior and resistance from the patients [18]. Therefore, aside from measures to overcome the barriers cited above, continuous and comprehensive communication training should also be established as studies showed that communication skills training increases the patients' satisfaction [20, 21].

Patients older than 65 years seem to be more worried if doctors explain things in a way they could understand. Cognitive decline can affect the information process and jeopardize the doctor-patient communication. Yet, more important is to consider that older patients often have more comorbidities which lead to more complex medical explanations and situations [22], which support the need for more comprehensive communication training.

The interaction between patients and doctors is likely to differ according to the patients' educational level [23], since this specific population has a higher focus in the emotional area of the consultation compared to patients with higher educational level [24]. The survey used in this study revealed an association between lower educational level (<4 years of college) and concerns regarding the respect of doctor's treatment (p = 0.027). It is proposed that lower educated patients feel more comfortable talking about the affective side of the doctors' relationship compared to higher educated people, which is more focused on the problem/treatment directions area [24]. Hence, the understanding of patient's needs are essential for delivering a high-quality healthcare.

Every year, around 700 thousand falls occur in US hospitals [25], with falling reported as one of the most common complications in hospitals [26]. Although there is no difference in the frequency of falls among genders [27], this study showed that women are more concerned about this aspect than men. Therefore, hospital initiatives to avoid falls in high-risk patients must be perceived as an institutional priority for a hospital to provide optimal patient care, especially by female patients [28].

A few patients (18.7%) searched hospital information in the internet prior to their visit. Among those, patients who had 4 years or more of college preferred Hospital Compare to obtain information about hospitals. Within this platform, more than 4,000 Medicare-certified hospitals in the US are rated in terms of quality of care, making it easier to compare and contrast hospitals. Federal agencies, accrediting organizations, employers, physicians, hospitals, and consumer organizations helped develop the system with the US Centers for Medicare & Medicaid Services [29]. The website presents information on parameters regularly measured by Medicare such as surgical complications, healthcare-associated infections and patient's experiences with each hospital. The presentation of some data includes graphics and bar charts, which may make it more attractive or understandable to individuals with more years of education who prefer to view information in the form of line and bar charts, as showed by this study. This preference may be due to their ability to better interpret information presented in this manner [30].

This study is subject to the usual limitations of cross-sectional studies. The sample size may not be broad enough to create generalizable information, so larger studies should further explore these specific aspects. One of the limitations of this study is that respondents with longer discharge dates from the hospital may have recall bias. However, the authors felt that what respondents felt were important to them would remain relatively constant compared to their actual ratings of their hospital stay. For example, respondents may rate their nursing care low or high depending on how they may recall their experience, but the level of importance they place on "nursing care," per se, as part of hospitalization, should stay relatively the same.

This study only includes participants in one hospital with a specific patient population. Results, therefore, may not apply to other hospitals. Although the missing data for each question is only less than two participants, the data should be interpreted with caution and should not be generalized. Moreover, patients would only respond to specific questions and were not able to write other concerns that were not included in the questionnaires. As a result, more in-depth studies could be done to focus on broader factors in hospitals. Lastly, physicians conducted surveys, which may have influenced the responses.

Another limitation of this study is that it was done prior to the COVID-19 pandemic. It is largely unknown how such a pandemic may affect what patients find important in terms of their hospital stays. However, the authors felt it would not be far-fetched to think that the following factors may be of heightened importance in patients' minds given this pandemic: 1) Caregivers in full personal protective equipment (PPEs) when appropriate; 2) Cleanliness measures they perceived in their rooms, such as more frequent cleanings, wiping; and 3) Witnessed handwashing behaviors in health care workers in the hospital. These considerations certainly deserve attention in future studies, especially given the current pandemic and the possibility that pandemics can recur.

The topics approached in the questionnaire are ultimately related to the patient experience during their visit to the hospital. A systematic review including 40 studies showed a positive association between the three domains of quality: clinical effectiveness, patient safety and patient experience. Although there is no causal effect among them, by analyzing the strengths and weakness of patient experience, as in this study, allow the provisioning of a better patient experience, which consequently will increase the likelihood of improvement in patient safety and clinical effectiveness, which are the pillar of quality in healthcare [31].

Surveys are routinely given to patients after receiving care in a hospital, but which aspects of care are most important to them has not been widely studied. Age, gender, and level of education showed to have an effect on how hospital care is perceived. Regardless of the characteristics of the population, the risk of getting an infection was the main concern overall, so it is important that hospitals promote actions to prevent it and share them with the patients. Knowledge of what different groups of patients prefer may assist hospital administrators in making institutional improvements.

## Supporting information

**S1 Appendix. Survey applied.**
(DOCX)

**S2 Appendix. Raw data collected.**
(DOCX)

## Acknowledgments

The authors would like to acknowledge Tara Brigham, Victoria Clifton, Zhuo Li, and Alison Dowdell, for their assistance in preparing this abstract.

## Presentations

Partial results of this study were presented at the 17th Annual Southern Hospital Medicine Conference (October 2016).

## Author Contributions

**Conceptualization:** George G. A. Pujalte, Isaac I. Effiong, Susannah S. Rothstein, Ryan Cudahy, Lorenzo O. Hernandez, Timothy M. Davlantes.

**Data curation:** George G. A. Pujalte, Tais G. O. Bertasi, Raphael A. O. Bertasi, Susannah S. Rothstein, Ryan Cudahy, Lorenzo O. Hernandez, Timothy M. Davlantes.

**Formal analysis:** George G. A. Pujalte, Isaac I. Effiong, Tais G. O. Bertasi, Raphael A. O. Bertasi, Susannah S. Rothstein, Ryan Cudahy, Lorenzo O. Hernandez, Timothy M. Davlantes.

**Investigation:** George G. A. Pujalte, Isaac I. Effiong, Tais G. O. Bertasi, Raphael A. O. Bertasi, Susannah S. Rothstein, Timothy M. Davlantes.

**Methodology:** George G. A. Pujalte, Isaac I. Effiong, Tais G. O. Bertasi, Raphael A. O. Bertasi, Susannah S. Rothstein, Ryan Cudahy, Lorenzo O. Hernandez, Timothy M. Davlantes.

**Project administration:** George G. A. Pujalte, Timothy M. Davlantes.

**Supervision:** George G. A. Pujalte, Timothy M. Davlantes.

**Validation:** George G. A. Pujalte, Isaac I. Effiong, Tais G. O. Bertasi, Raphael A. O. Bertasi, Susannah S. Rothstein, Ryan Cudahy, Lorenzo O. Hernandez, Timothy M. Davlantes.

**Visualization:** Isaac I. Effiong, Tais G. O. Bertasi, Raphael A. O. Bertasi, Susannah S. Rothstein, Ryan Cudahy, Lorenzo O. Hernandez, Timothy M. Davlantes.

**Writing – original draft:** George G. A. Pujalte, Isaac I. Effiong, Tais G. O. Bertasi, Raphael A. O. Bertasi, Susannah S. Rothstein, Ryan Cudahy, Lorenzo O. Hernandez, Timothy M. Davlantes.

**Writing – review & editing:** George G. A. Pujalte, Isaac I. Effiong, Tais G. O. Bertasi, Raphael A. O. Bertasi, Ryan Cudahy, Lorenzo O. Hernandez, Timothy M. Davlantes.

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
