## [Decision Letter · Decision Letter 0]

8 Mar 2021

PONE-D-20-24334

Do Patients’ Characteristics Influence Their Healthcare Concerns? – A Hospital Care Survey

PLOS ONE

Dear Dr.Tais Garcia de Oliveira Bertasi,

Thank you for submitting your manuscript to PLOS ONE. After careful consideration, we feel that it has merit but does not fully meet PLOS ONE’s publication criteria as it currently stands. Therefore, we invite you to submit a revised version of the manuscript that addresses the points raised during the review process.

We look forward to receiving your revised manuscript.

Kind regards,

Sharon Mary Brownie

Academic Editor

PLOS ONE

Journal Requirements:

Additional Editor Comments 

Reviewers have offered a significant number of recommendations to strengthen your paper. Please respond in full to each suggestion offered

Reviewers' comments:

Reviewer's Responses to Questions

**Comments to the Author**

1. Is the manuscript technically sound, and do the data support the conclusions?

Reviewer #1: Yes

Reviewer #2: Partly

Reviewer #3: Partly

2. Has the statistical analysis been performed appropriately and rigorously? 

Reviewer #1: Yes

Reviewer #2: I Don't Know

Reviewer #3: Yes

3. Have the authors made all data underlying the findings in their manuscript fully available?

Reviewer #1: Yes

Reviewer #2: No

Reviewer #3: No

4. Is the manuscript presented in an intelligible fashion and written in standard English?

Reviewer #1: Yes

Reviewer #2: Yes

Reviewer #3: Yes

5. Review Comments to the Author

Reviewer #1: 1. if a linelist of the potential samples were made to select study samples at random.

2. could all the tables and figures have a title on the top of each.

3. could there be chance of recall bias particularly for aged respondents while the interview was done 50 or more days after discharge from the hospital.

4. The demographic information table (table 1) could contain information of distribution of respondents by gender and age group.

Reviewer #2: It is not possible to interpret the findings in the absence of information on the randomization process and the response rate to the survey, both overall and in terms of key characteristics of the repondents

Reviewer #3: This is an interesting study, but there are some concerns that need to be addressed.

After reading the introduction, I was unsure whether the authors disagree with using the CAHPS for reporting hospital quality or not. Although the CAHPS/CMS aggregates measures for hospital reporting (e.g., hospital compare), researchers could request the data to understand more granular level information and patterns (e.g., patient characteristics). Link: https://hcahpsonline.org/globalassets/hcahps/survey-instruments/mail/qag-v16.0-materials/2021_survey-instruments_english_mail.pdf

I would encourage the authors to review sections of the introduction. For example: “Some factors that affect patient survey scores may not be directly related to hospital performance. [5] Characteristics of different patient populations may make certain parts of the survey more relevant than others. Hospitals may have very little control on the education level and age of patients who present for care. It is known that more educated and younger patients tend to evaluate health care less positively.” I would hope that hospitals “want” to treat everyone equally. Surveys are designed to get a range of responses that would be representative of the overall population.

The authors mentioned “A survey developed via literature review and specialist recommendations was applied in different departments of our tertiary care center during the year of 2016.” More information about how the survey questions were developed is needed. What article(s) or literature was used to develop the questions?

It is unclear to me why the patients needed to be recently admitted to the hospital or admitted more than 50 days before the survey was being applied. Explaining this range may be important. In addition, what does it mean to be recently admitted?

More details are needed about “The investigators randomly approached the patients in different hospital departments to solicit their participation.” Are these the authors in the paper, RAs, others? How many investigators were conducting the surveys? Why were participants approached randomly? I would assume that the investigators were hoping for some representation. How many people declined to participate?

In the discussion, the authors mentioned that “‘Acquiring an infection while receiving health care is also known as “health care-associated Infections” (HAI) [11]’” They may need to expand about how common these infections are and provide some explanations as to why they believe this was the top concern.

The authors also mentioned that “physicians conducted the surveys, which may have influenced the responses.” How do the authors believe that this may have influenced the responses?

Finally, these data are somewhat dated. The authors may need to include a paragraph in the discussion about how the current epidemic may have changed/influences or make these survey more relevant.

6. PLOS authors have the option to publish the peer review history of their article (what does this mean?). If published, this will include your full peer review and any attached files.

Reviewer #1: **Yes: **Dr. Md. Ziaur Rahman

Reviewer #2: No

Reviewer #3: No

---

## [Author Response · Author response to Decision Letter 0]

28 Jul 2021

Reviewer's Responses to Questions

1. Is the manuscript technically sound, and do the data support the conclusions?

Reviewer #1: Yes

Reviewer #2: Partly

Reviewer #3: Partly

2. Has the statistical analysis been performed appropriately and rigorously?

Reviewer #1: Yes

Reviewer #2: I Don't Know

Reviewer #3: Yes

3. Have the authors made all data underlying the findings in their manuscript fully available?

Reviewer #1: Yes

Reviewer #2: No

Reviewer #3: No

4. Is the manuscript presented in an intelligible fashion and written in standard English?

Reviewer #1: Yes

Reviewer #2: Yes

Reviewer #3: Yes

The authors appreciate the reviewers’ availability and great suggestions. We made several edits to the manuscript and included Appendix 2 with the raw data collected with the survey answers. 

Reviewer's Comments to the Author

REVIEWER 1:

1. If a linelist of the potential samples were made to select study samples at random.

The authors appreciate your comment. The randomization process was included in the Methods section and is described below in the answer for the 1st comment from Reviewer 2. 

2. Could all the tables and figures have a title on the top of each.

Thank you for your suggestion. We included a title on the top of the figures and table.

3. Could there be chance of recall bias particularly for aged respondents while the interview was done 50 or more days after discharge from the hospital.

The authors appreciate your comment. We have indicated this possibility in the limitations section.

4. The demographic information table (table 1) could contain information of distribution of respondents by gender and age group.

Thank you for your suggestion. We added the age and sex information on Table 1. 

REVIEWER 2:

1. It is not possible to interpret the findings in the absence of information on the randomization process and the response rate to the survey, both overall and in terms of key characteristics of the respondents

The authors appreciate your comment. The randomization process was included in the Methods section. The authors of the study were the investigators who approached patients in waiting rooms in all the Mayo buildings, in hospital rooms, and patient rooms, depending on the setting they are in, at various days of the study period. They asked if patients were recently hospitalized if they were in the waiting room or patient room. A total of 142 patients declined participating in the study, with a response rate of 64.8%. Consent was obtained for those interested and study survey was given.

Participants were approached randomly to maximize experience breadth. For example, if only patients in one building were approached, only patients admitted into specific services (e.g. Family Medicine, Internal Medicine, Rehabilitation Medicine) would be surveyed. The authors wanted to cover all experiences from admission into all services offered in the hospital, as much as possible and hoped that by spreading out in this manner, there would be good representation of all types of patient admission.

REVIEWER 3:

This is an interesting study, but there are some concerns that need to be addressed.

1. After reading the introduction, I was unsure whether the authors disagree with using the CAHPS for reporting hospital quality or not. Although the CAHPS/CMS aggregates measures for hospital reporting (e.g., hospital compare), researchers could request the data to understand more granular level information and patterns (e.g., patient characteristics). Link: https://hcahpsonline.org/globalassets/hcahps/survey-instruments/mail/qag-v16.0-materials/2021_survey-instruments_english_mail.pdf

Thank you for your comment and for the link. The authors agree that it was not clear whether we disagree with using the CAHPS for reporting hospital quality. Therefore, we included some sentences at the end of the Introduction section to make it clearer and to explain that the aim of our study is to understand in more detailed the patients’ concerns behind each response to those surveys.

2. I would encourage the authors to review sections of the introduction. For example: “Some factors that affect patient survey scores may not be directly related to hospital performance. [5] Characteristics of different patient populations may make certain parts of the survey more relevant than others. Hospitals may have very little control on the education level and age of patients who present for care. It is known that more educated and younger patients tend to evaluate health care less positively.” I would hope that hospitals “want” to treat everyone equally. Surveys are designed to get a range of responses that would be representative of the overall population.

The authors appreciate your comment. We have changed the introduction to make our message clearer.

3. The authors mentioned “A survey developed via literature review and specialist recommendations was applied in different departments of our tertiary care center during the year of 2016.” More information about how the survey questions were developed is needed. What article(s) or literature was used to develop the questions?

Thank you for your suggestion. We included the references of the literature used to develop the survey. 

4. It is unclear to me why the patients needed to be recently admitted to the hospital or admitted more than 50 days before the survey was being applied. Explaining this range may be important. In addition, what does it mean to be recently admitted?

Thank you for your comment. Recently admitted means admitted within 1-2 weeks. The survey from Mayo Clinic usually arrives more than 50 days after patients are admitted. The reason for this is unclear but may be because surveyors want to determine what remains in patients’ recollections after such a period of time has elapsed.

5. More details are needed about “The investigators randomly approached the patients in different hospital departments to solicit their participation.” Are these the authors in the paper, RAs, others? How many investigators were conducting the surveys? Why were participants approached randomly? I would assume that the investigators were hoping for some representation. How many people declined to participate?

The authors of the study were the investigators who approached patients in waiting rooms in all the Mayo buildings, in hospital rooms, and patient rooms, depending on the setting they are in, at various days of the study period. They asked if patients were recently hospitalized if they were in the waiting room or patient room. Consent was obtained for those interested and study survey was given.

Participants were approached randomly to maximize experience breadth. For example, if only patients in one building were approached, only patients admitted into specific services (e.g. Family Medicine, Internal Medicine, Rehabilitation Medicine) would be surveyed. The authors wanted to cover all experiences from admission into all services offered in the hospital, as much as possible and hoped that by spreading out in this manner, there would be good representation of all types of patient admission. A sentence was included in the Methods section to make it clearer. A total of 142 patients declined participating in the study (the response rate was included in the Results section)

6. In the discussion, the authors mentioned that “‘Acquiring an infection while receiving health care is also known as “health care-associated Infections” (HAI) [11]’” They may need to expand about how common these infections are and provide some explanations as to why they believe this was the top concern.

Thank you for your suggestion. The authors included a few sentences in the Discussion section about the frequency of HAIs and why we believe this was a top concern for the patients. 

7. The authors also mentioned that “physicians conducted the surveys, which may have influenced the responses.” How do the authors believe that this may have influenced the responses?

The authors believe that having physicians conducting the surveys may have influenced who they gave surveys to. For example, physicians may not want to give the surveys to patients they took care of. However, the authors/investigators tried not to be affected by this factor when giving out the surveys.

8. Finally, these data are somewhat dated. The authors may need to include a paragraph in the discussion about how the current epidemic may have changed/influences or make this survey more relevant.

The authors appreciate this interesting comment. We included a comment about this in the discussion section.

---

## [Decision Letter · Decision Letter 1]

18 Aug 2021

PONE-D-20-24334R1

Do Patients’ Characteristics Influence Their Healthcare Concerns? – A Hospital Care Survey

PLOS ONE

Dear Dr.Tais Garcia de Oliveira Bertasi,

Thank you for submitting your manuscript to PLOS ONE. After careful consideration, we feel that it has merit but does not fully meet PLOS ONE’s publication criteria as it currently stands. Therefore, we invite you to submit a revised version of the manuscript that addresses the points raised during the review process.

We look forward to receiving your revised manuscript.

Kind regards,

Sharon Mary Brownie

Academic Editor

PLOS ONE

Journal Requirements:

Editor Comments 

Reviewers have requested some additional information and improvements in your methods section. Please pay careful attention to what is requested and respond appropriately.

Reviewers' comments:

Reviewer's Responses to Questions

**Comments to the Author**

1. If the authors have adequately addressed your comments raised in a previous round of review and you feel that this manuscript is now acceptable for publication, you may indicate that here to bypass the “Comments to the Author” section, enter your conflict of interest statement in the “Confidential to Editor” section, and submit your "Accept" recommendation.

Reviewer #1: All comments have been addressed

Reviewer #2: (No Response)

2. Is the manuscript technically sound, and do the data support the conclusions?

Reviewer #1: Yes

Reviewer #2: Partly

3. Has the statistical analysis been performed appropriately and rigorously? 

Reviewer #1: Yes

Reviewer #2: I Don't Know

4. Have the authors made all data underlying the findings in their manuscript fully available?

Reviewer #1: Yes

Reviewer #2: No

5. Is the manuscript presented in an intelligible fashion and written in standard English?

Reviewer #1: Yes

Reviewer #2: Yes

6. Review Comments to the Author

Reviewer #1: The respondent selection criteria regarding hospital admission as mentioned."recently admitted or admitted more the 50 days before the survey was applied" is not clear. If data was collected from patients admitted a week before or 90 days before data collection. Should it not be a range e.g. admitted 10 - 50 days before data collection.

Reviewer #2: This version does not address my previous comments. Simply stating that patients were approached randomly does not adequately describe the randomisation process. Further, the response rate according to key characteristics is not provided. It could be that the response rate varied significantly by race, education, age gender etc, so introducing potential biases. This information is required for a determination of the validity of the findings

7. PLOS authors have the option to publish the peer review history of their article (what does this mean?). If published, this will include your full peer review and any attached files.

Reviewer #1: **Yes: **Dr. Md. Ziaur Rahman, Epidemiologist and Public Health Specialist

Reviewer #2: **Yes: **Prof Ian Ring

---

## [Author Response · Author response to Decision Letter 1]

23 Sep 2021

REVIEWER #1: 

The respondent selection criteria regarding hospital admission as mentioned, "recently admitted or admitted more the 50 days before the survey was applied" is not clear. If data was collected from patients admitted a week before or 90 days before data collection, should it not be a range (e.g. admitted 10 - 50 days before data collection)?

The authors agree with your comment. The methodology section was rewritten in agreement with your recommendation. 

REVIEWER #2: 

This version does not address my previous comments. Simply stating that patients were approached randomly does not adequately describe the randomisation process. Further, the response rate according to key characteristics is not provided. It could be that the response rate varied significantly by race, education, age gender, etc., so introducing potential biases. This information is required for a determination of the validity of the findings. 

The authors appreciate your comment. The methodology section was rewritten to address the previous comment. Regarding the response rate, as it is possible to see on the appendix showing all data, the maximum of missing answers for each of the nine questionnaire questions were from two participants; this fortunately would not make a statistically significant difference when accounting for race, education, age, and gender. However, the reviewer made a great point, and the potential bias is now included in the limitations section.

---

## [Editor Report · Decision Letter 2]

4 Oct 2021

Do Patients’ Characteristics Influence Their Healthcare Concerns? – A Hospital Care Survey

PONE-D-20-24334R2

Dear Dr. Tais Garcia de Oliveira Bertasi,

We’re pleased to inform you that your manuscript has been judged scientifically suitable for publication and will be formally accepted for publication once it meets all outstanding technical requirements.

Kind regards,

Sharon Mary Brownie

Academic Editor

PLOS ONE

Editor Comments 
---

## [Editor Report · Acceptance letter]

7 Oct 2021

PONE-D-20-24334R2 

Do Patients’ Characteristics Influence Their Healthcare Concerns? – A Hospital Care Survey 

Dear Dr. Bertasi:

I'm pleased to inform you that your manuscript has been deemed suitable for publication in PLOS ONE. Congratulations! Your manuscript is now with our production department. 

Kind regards, 

on behalf of

Professor Sharon Mary Brownie 

Academic Editor

PLOS ONE